# The Effects of Radiotherapy on the Sequence and Eligibility of Breast Reconstruction: Current Evidence and Controversy

**DOI:** 10.3390/cancers16172939

**Published:** 2024-08-23

**Authors:** Andrew R. Campbell, Alexander J. Didier, Taha M. Sheikh, Sami Ansari, Dean E. Watkins, Alan M. Fahoury, Swamroop V. Nandwani, Mohammad Rashid

**Affiliations:** Department of Medicine, College of Medicine and Life Sciences, The University of Toledo, Toledo, OH 43606, USA; alexander.didier@rockets.utoledo.edu (A.J.D.); dr.tahasheikh@gmail.com (T.M.S.); sami.ansari@vumc.org (S.A.); dean.watkins@rockets.utoledo.edu (D.E.W.); alan.fahoury@rockets.utoledo.edu (A.M.F.); swamroop.nandwani@rockets.utoledo.edu (S.V.N.); mohammad.rashid@utoledo.edu (M.R.)

**Keywords:** breast cancer, breast reconstruction, radiotherapy, quality of life, mastectomy

## Abstract

**Simple Summary:**

Breast reconstruction is considered for any patient who is undergoing surgical treatment for breast cancer. Many factors must be considered when discussing reconstruction, including patient risk factors, the type of material used for breast restoration, and the timing of surgery relative to other adjuvant treatments, such as radiotherapy. Plans for post-mastectomy radiotherapy significantly impact whether a breast reconstruction may be performed at the same time as a mastectomy (immediate reconstruction) or delayed for a later time (delayed reconstruction). The risk of surgical complications from irradiating a reconstructed breast is often weighed against the benefits of fewer surgeries and the potential for an improved quality of life when discussing eligibility for immediate reconstruction. This narrative review covers the current trends and controversies regarding breast reconstruction and radiotherapy. We aim to provide the context and relevant evidence for general community-practice oncologists and oncology providers who counsel patients on the sequencing and modalities of radiotherapy and breast reconstruction.

**Abstract:**

Immediate breast reconstruction (IBR) following a mastectomy, combined with radiotherapy, presents a multifaceted approach to breast cancer treatment, balancing oncological safety and aesthetic outcomes. IBR, typically involving the use of implants or autologous tissue, aims to restore breast morphology directly after a mastectomy, minimizing the psychological and physical impacts. However, integrating radiotherapy with IBR is complex due to the potential adverse effects on reconstructed tissues. Radiotherapy, essential for reducing local recurrence, can induce fibrosis, capsular contracture, and compromised aesthetic results. This narrative review covers the current trends in the sequencing of breast reconstruction and radiotherapy. We discuss patient selection, timing of radiotherapy, and reconstructive techniques, with special attention paid to quality-of-life outcomes that are increasingly reported in clinical trials. Emerging evidence supports the feasibility of IBR with careful patient selection and tailored therapeutic approaches, although ongoing research is necessary to refine protocols and enhance outcomes. Overall, IBR in the context of radiotherapy remains a promising but intricate treatment modality, requiring a nuanced balance between cancer control and aesthetic restoration.

## 1. Introduction

The landscape of breast cancer treatment has evolved significantly over the years, with an increasing emphasis on not only eradicating the disease, but also prioritizing the holistic well-being and quality of life of affected individuals. Immediate breast reconstruction (IBR) after mastectomy has emerged as a crucial component in this paradigm shift. This approach involves the reconstruction of the breast immediately after a mastectomy in the same operating room, mitigating the need for a second surgery [1]. Various techniques, such as implants, autologous tissue transfer, or a combination of both, may be employed in immediate breast reconstruction. Candidates for IBR are selected based on individual patient characteristics and preferences, stage of the disease, and need for adjuvant therapies [1].

As it currently stands, the rate of post-mastectomy breast reconstruction is low, with significant differences among rural and urban facilities in the United States and across other developed nations. However, the most important consideration before choosing immediate versus delayed surgery is whether the patient may need post-mastectomy radiotherapy (PMRT). As the indications for adjuvant radiation expand to more patients, the discussion regarding safely performing immediate surgery is more relevant than ever. Algorithms that determine the optimal reconstruction strategy predominantly focus on avoiding radiation toxicity to a definitive reconstruction. Current strategies include staging reconstruction with the placement of a tissue expander, delaying reconstruction 6–12 months after PMRT, or foregoing reconstruction entirely. Historically, the need for radiation has precluded eligibility for IBR, but the incorporation of PMRT after immediate reconstruction is rising, even among patients who were previously considered high risk for surgical complications. The sequence of reconstructive surgery and radiotherapy remains a hotly debated issue.

In this review, we will provide an overview of the current body of knowledge surrounding the trends and controversies of radiotherapy and breast reconstruction, focusing on the epidemiology and clinicopathological characteristics of eligible patients. Our review will highlight the implications of IBR on the timing of adjuvant radiotherapy and clinical considerations for patients with past radiation exposure.

## 2. Breast Reconstruction Techniques

After proceeding with the mastectomy, patients have several options for reconstructive surgery (Figure 1). While delayed breast reconstruction (DBR) surgery was formerly preferred for most patients, immediate breast reconstruction (IBR)—occurring under the same anesthetic as the mastectomy—has become increasingly popular in recent years. Immediate reconstruction offers instant physical restoration, fewer trips to the operating room, and reduced short-term costs. Clinically, IBR is becoming more feasible due to improved surgical techniques that preserve more healthy, adequately vascularized tissue while also reducing the frequency of possible complications. This is partly thanks to the advancement of conservative mastectomy techniques—skin-sparing mastectomies and nipple-sparing mastectomies—which provide patients the opportunity to pursue higher-quality reconstructions compared to conventional mastectomies as more original breast tissue is spared. Due to the nature of these techniques, there is a possible increased risk for cancerous tissue to be left behind, which is why positive intra-operative margins are contraindications for nipple-sparing mastectomies [2]. Additionally, these conservative techniques have a high risk for tissue necrosis due to reduced vascularity [2]. Importantly, the post-reconstruction complication rates in patients who receive skin-sparing and nipple-sparing mastectomies are comparable to those of non-conservative mastectomies [2].

Post-mastectomy reconstruction modalities include single-stage direct-to-implant, two-stage (tissue expander followed by implants), and autologous reconstructions. Autologous reconstruction can be performed immediately or delayed (with or without a tissue expander) and can be further classified by either the use of vascularized tissue or non-vascularized fat tissue. Patients require an adequate surface area of flap tissue or volume of body fat to be eligible for autologous reconstruction. For both IBR and DBR, implant-based reconstruction has been most commonly performed with implant placement sub-muscularly underneath the pectoralis major, with more recent techniques permitting the placement of implants pre-pectorally [3].

### 2.1. Overview of Immediate Reconstruction Options

In single-stage direct-to-implant reconstructions, a breast implant is immediately placed in the breast at the time of completion of a skin-sparing mastectomy. This technique is ideal for patients who do not require PMRT, as it bypasses a delay in achieving the desired breast aesthetic with no risk of radiation-associated complications during the healing process [4].

Patients may also be considered for an immediate autologous reconstruction rather than an implant-based reconstruction. Autologous breast reconstruction can be categorized using vascularized or non-vascularized tissue for grafts. Vascularized autologous tissue for breast reconstruction is commonly harvested from the thigh, abdomen (transverse rectus abdominis myocutaneous (TRAM) flap and Deep Inferior Epigastric Artery Perforator (DIEP) flap), back (latissimus dorsi (LD) flap), or gluteal region (superior and inferior gluteal artery perforator (S-GAP and I-GAP) flaps), and then most commonly anastomosed to the internal mammary artery and vein [5,6]. Alternatively, non-vascularized autologous lipoaspirate grafts utilize tissue harvested via liposuction from various donor sites, which is then introduced into the breast in small volumes using a multi-planar approach for further correction during multiple rounds of grafting over the course of 12–18 months [5]. Such lipoaspirate grafts can be used in conjunction with implants and spacers for further corrective management in a technique known as lipofilling.

Each reconstruction technique offers its own strengths and weaknesses (Table 1). Autologous tissue-based reconstruction comes with its own risks and benefits. One major weakness, particularly when compared to implant-based IBR, is the time it takes to perform a complete reconstruction. In the case of vascularized autologous tissue reconstruction, this refers to a longer duration of surgery, whereas for non-vascular autologous lipoaspirate grafting, reconstruction occurs over the span of several months with multiple rounds of follow-up visits [5]. There is no significant difference between the frequency of reoperations or the frequency of immediate post-operative complications (such as bleeding, seroma, or delayed wound healing) between autologous and implant-based IBR [5,7]. However, patients who undergo autologous tissue-based reconstructions are significantly less likely to suffer major complications compared to patients who receive implant-based reconstructions, even prior to PMRT [8,9]. For example, there is a significantly higher rate of implant loss for patients who receive radiotherapy after IBR with implants if radiotherapy is needed [7]. However, this approach remains attractive to patients as it offers them the fastest -route to reconstructed breasts with the least amount of time in the operating room.

### 2.2. Overview of Delayed Reconstruction Options

Two-stage reconstruction using expander implants after a mastectomy can provide similarly optimal results for patients who have an indeterminate need for PMRT [10]. This technique utilizes a two-step process, in which a patient first undergoes a skin-sparing mastectomy and the placement of a tissue expander in the breast. If the patient does not require PMRT after surgical staging, they then receive a reconstruction at any time, whereas upstaged patients undergo PMRT and a standard delayed reconstruction is performed [10]. Using an expander implant up front allows more time for patients to decide between final reconstruction modality and revision.

Delayed autologous reconstruction involves a similar technique to immediate autologous reconstruction and offers the same surgical and cosmetic benefits; however, it occurs over a longer timeframe and may require multiple operations. To be a candidate for autologous tissue graft reconstruction, patients require an adequate surface area of highly vascularized flap tissue. This is because inadequate graft vascularization increases the risk of post-operative complications, including wound infection, flap necrosis, and poor wound healing, which can either lead to graft failure and the delay of PMRT administration, or breast deformity after PMRT [7,11]. In the past, non-vascularized lipoaspirate grafts have been used alone for reconstruction, however this technique has fallen out of favor due to increased risks of graft failure due to the effects of radiation therapy on large volumes of poorly perfused tissue.

Delayed autologous reconstruction and two-stage expander implant reconstruction each have their own advantages and disadvantages. Delayed autologous implants may be more cost-effective in the long term due to a lower rate of wound contracture, fat necrosis, and revision surgeries. However, initially, the upfront cost of performing an autologous reconstruction is higher. Use of the expander implant allows for revisions of asymmetries or the effects of radiation on the tissue. Also, if the patient uses an expander, they do not necessarily have to use an implant and may switch to autologous reconstruction at a later date, giving them the flexibility to choose between both modalities. On the other hand, using a two-stage expander involves a relatively higher risk of complications compared to delayed autologous implants, including problems with the tissue expander in smokers and with radiation therapy. These complications include infection, skin breakdown, and capsular contracture, even if PMRT is performed before the implant is inserted.

## 3. Influence of Post-Mastectomy Radiotherapy on Choice of Reconstruction

The decision to undergo immediate vs. delayed reconstruction is heavily influenced by the need for radiotherapy after a mastectomy. However, the surgical pathology often dictates the need for radiotherapy, which complicates decision making. While IBR possesses many advantages, delayed reconstruction offers the patient a safe and flexible avenue to receive radiotherapy before definitive reconstruction. There is no risk of delaying PMRT initiation due to reconstruction failure and delivery of radiation is simpler on the more two-dimensional chest wall compared to a three-dimensional reconstructed breast [12]. Importantly, patients who receive IBR will not receive final pathology results at the time of reconstruction, which complicates the accurate delivery of PMRT. This also may be a contributing factor to the increased frequency of late complications associated with PMRT use due to the irradiation of either the implant and implant-adjacent tissue, or the autologous graft tissue [12].

Previous evidence suggested radiation to autologous flaps carries excessive risks, with unacceptable rates of fat necrosis (44% vs. 9%), tissue fibrosis and contracture (75% vs. 0%), and breast volume loss (88% vs. 0%) [12]. Unfortunately, some of these potentially severe complications then ultimately delay successive rounds of radiation therapy for patients. More recent prospective comparisons suggest autologous flaps may tolerate radiotherapy better than previously shown results, however this risk causes many surgeons to prefer implant-based reconstructions if IBR is desired and radiation may be indicated. Despite this, patients who underwent immediate autologous reconstructions prior to PMRT were observed to be less likely to require a re-operation due to complications when compared to patients undergoing delayed autologous reconstructions (20% vs. 32%), and were also found to have a similar frequency of reconstruction failure (4% vs. 5%) [12].

Both immediate implant and two-stage expander reconstructions have higher rates of complications compared to autologous reconstructions if PMRT is required. For patients who received a two-stage expander reconstruction, PMRT is associated with a high risk of implant failure, but lower rates of capsular contracture and improved aesthetic outcomes compared with immediate implant reconstruction. Cordeiro et al. found no significant difference between the frequency of reconstruction failure between implant-based and tissue-spacer reconstructions, despite predicting a more significant 6-year failure rate favoring implant-based reconstructions (32% vs. 16.4%; *p* < 0.01) [13]. In another study, 20% of patients (n = 95) receiving two-stage expander reconstructions prior to PMRT experienced reconstruction failure at the two-year follow up [14]. Cordeiro’s study observed that patients who received implant-based reconstructions were more likely to develop grade 3 (44.6% vs. 15.9%; *p* < 0.01) and grade 4 (6.3% vs. 1.22%; *p* < 0.01) capsular contractures compared to those who received tissue expander reconstructions [13]. Radiotherapy appears to affect the aesthetics and integrity of the implant, regardless of timing.

The choice of reconstruction modality can also influence the delivery of radiotherapy. Motwani et al. [15]. investigated how IBR impacted the delivery of PMRT by showing how a reconstructed breast is more difficult to irradiate and may compromise efficacy. In this study, the severity of the deviation from optimal treatment conditions for each patient (n = 110) was determined and scored [15]. Then, it was compared between patients who underwent immediate reconstructions and those who did not. The optimal treatment parameters were based on the degree of chest wall coverage, treatment of the ipsilateral internal mammary chain, and minimizing radiation to both the heart and lungs. Patients who did not undergo breast reconstructions prior to PMRT saw significantly better coverage of the chest wall (100% vs. 79%; *p* < 0.0001) and treatment delivery to the ipsilateral internal mammary chain (93% vs. 45%; *p* < 0.0001) [15]. Over half of patients in this study who received IBR had compromised PMRT regimen optimizations compared to just 7% of those who did not undergo reconstructions (*p* < 0.0001) [15]. The results of this study display a clear hinderance of optimal PMRT administration to patients receiving IBR. While improvements to the PMRT technique have been developed since this study, there is a continued discussion on the safe delivery of radiation after immediate reconstruction, including the use of lipofilling. While this technique has been shown to improve breast volume and symmetry in DBR patients after PMRT, poor reconstruction outcomes have been observed in patients who received neoadjuvant radiotherapy prior to a mastectomy and subsequent reconstruction [16,17].

## 4. Complications

The evaluation of radiation-related complications in reconstructed breasts is a critical aspect of post-mastectomy care, regardless of reconstruction modality. Radiation therapy is often employed as an adjuvant treatment following a mastectomy to reduce the risk of local recurrence; however, it can significantly impact the outcome of breast reconstruction. Additionally, the timing of radiotherapy may influence early complication rates (surgical site infections, flap necrosis, and seroma/hematoma) and late complication rates (capsular contracture, need for revision surgery, and reconstructive failure) (Table 2).

**Table 2 cancers-16-02939-t002:** Comparison of complication rates among different breast reconstruction modalities in the setting of PMRT.

Author	Study Design	Patients (n)	Reconstruction Modality	Complication Rates
Tran et al. [18].	Retrospective	41	Immediate autologous	Fat necrosis (34%) and asymmetry (78%)
Billig et al. [19].	Prospective	108	Immediate autologous	Fat necrosis (16.9%) and hematoma (5.6%)
Billig et al. [19].	Prospective	67	Delayed autologous	Fat necrosis (19.5%) and wound dehiscence (9%)
Albino et al. [20].	Retrospective	76	Immediate autologous	Fat necrosis (19%) and contracture or scarring (30%)
Dewael et al. [21].	Retrospective	20	Immediate autologous	Fat necrosis (60%), contracture (60%), and infection (20%)
Dewael et al. [21].	Retrospective	40	Delayed autologous	Fat necrosis (12%) and wound dehiscence (10%)
Jhaveri et al. [22].	Retrospective	69	Immediate autologous	Grade 2–4 complications (55%)
Jhaveri et al. [22].	Retrospective	23	Delayed implant	Grade 2–4 complications (8%)
Maalouf et al. [23].	Retrospective	30	Immediate autologous	Reoperation (40%)
Maalouf et al. [23].	Retrospective	32	Delayed autologous	Reoperation (12%)
McCarthy et al. [24].	Prospective	10	Delayed implant	Capsular contracture (60%)
Cordeiro et al. [25].	Retrospective	68	Delayed implant	Capsular contracture (68%)
Rella et al. [26].	Retrospective	80	Delayed implant	Capsular contracture (15%) and seroma (10%)
Benediktsson et al. [27].	Prospective	24	Immediate implant	Capsular contracture (41%)

Recent retrospective studies have assessed the complication rates between PMRT and breast reconstruction and found no differences between immediate vs. delayed reconstructions. Ogita and colleagues performed a retrospective cohort analysis of 81 patients treated with radiotherapy to immediate tissue expander or permanent implant-based reconstructions to assess for complications and risk factors. They found that the rates of total reconstruction failure, re-operation, and infection were 12.3%, 13.6%, and 11.1%, respectively. No significant difference in the rates of complications between the two groups were discovered. Age ≥ 55 years was statistically significant for re-operation (HR = 4.64, 95% CI: 1.27–16.9, *p* = 0.02) and infection (HR = 4.6, 95% CI 1.08–19.5, o = 0.04). However, no significant differences were observed for reconstruction failure, re-operation, or infection rates based on BMI ≥ 25, smoking history, clinical stage, and receipt of chemotherapy (neoadjuvant or adjuvant) [28]. Another retrospective cohort study compared complication rates in women who underwent immediate breast reconstructions and received PMRT to either permanent implants or temporary tissue expanders. The cohort consisted of 29 women who received PMRT to implants and 14 who received PMRT to tissue expanders [29]. Complication rates were similar between groups for superficial wound infection (3.4% vs. 7.1%), periprosthetic infection (3.4% vs. 7.1%), capsular contracture (41.4% vs. 21.4%), revision surgery for aesthetics (41.4% vs. 21.4%), wound dehiscence and device exposure (3.4% vs. 21.3%), and reconstructive failure (10.3% vs. 6.7%). Total complication rates were similar between groups (51.7% vs. 42.9%). Similarly, a retrospective cohort study evaluating 36 immediate and 89 delayed autologous reconstructions receiving PMRT found similar overall complication rates, surgical site infections, and fat necrosis between groups [30]. Revision rates were significantly lower in the immediate reconstruction group.

However, pooled analyses have shown higher rates of complications with immediate reconstruction. A meta-analysis of 30 studies comparing complication rates in immediate and delayed breast reconstructions showed higher rates of surgical complications (OR 1.30, 95% CI 1.03, 1.65; *p* = 0.03), infection (OR 1.41, 95% 1.04, 1.92; *p* = 0.03) and hematoma/seroma (OR 2.01, 95% CI 1.27–3.17; *p* = 0.003) in women who received IBR [31]. Additionally, there were no significant differences in outcomes in patients who received radiotherapy compared with those who did not. Another meta-analysis of 44 studies, including 3473 patients who underwent immediate or delayed autologous breast reconstructions in the setting of PMRT, showed significantly higher rates of fat necrosis in the immediate reconstruction group compared with delayed reconstruction (14.91% vs. 8.12%, *p* = 0.076) [32]. Rates of flap loss, hematoma, infection, and thrombosis were similar between groups, however seroma rates were lower in the immediate reconstruction group (2.69% vs. 10.57%, *p* = 0.042).

The optimal timing of PMRT and reconstructive type has yet to be determined, and avoidance of complications plays a significant role in deciding on a treatment sequence. A meta-analysis of 16 studies, including 2322 breast reconstructions, sought to answer this question [33]. This study found that autologous reconstruction following PMRT was most effective for avoiding complications compared with implant-based reconstructive strategies (OR = 0.10, 95% CI 0.02–0.55). This sequence was particularly effective at reducing infection rates, demonstrating a significant improvement compared with PMRT followed by tissue expanders/implants (OR = 0.12, 95% CI 0.02 to 0.88). A subgroup analysis of expander/implant reconstructions found that PMRT after placement reduced failure rates (OR = 0.35, 95% CI 0.15–0.81) compared to other implant-based strategies but increased the rates of capsular contracture. Another meta-analysis of seven studies, including 2921 patients undergoing one- and two-stage implant reconstructions, highlights the well-known risk of PMRT on implant complications, showing a significant increase in capsular contracture (OR 10.21, 95% CI 3.74 to 27.89, *p* < 0.00001), need for revisional surgery (OR 2.18, 95% CI 1.33 to 3.57, *p* = 0.002), and reconstructive failure (OR 2.52, 95% CI 1.48 to 4.29, *p* < 0.0007) [34]. Thus, there is much debate in the current literature regarding the timing of PMRT and reconstruction, as well as the differences between reconstruction modalities. Clinicians must bear in mind individual patient risk factors for complications and weigh these against the potential for success.

## 5. Influence of Post-Mastectomy Radiotherapy on Survival Outcomes

There is a significant debate regarding the optimal sequence of PMRT and reconstructive surgery, and there is still no level 1 evidence suggesting an appropriate treatment strategy. It is clear, however, that PMRT is supported by DFS and OS benefits in node-positive disease with four or more positive axillary lymph nodes (ALNs) after a mastectomy and ALN dissection. There is growing evidence for PMRT in node-negative disease with certain high-risk features as well. A patient’s need for PMRT may be influenced by local recurrence rates, survival benefits, and potential for harm to the patient.

### Impact of Nodal and Molecular Subtypes on Locoregional Recurrence (LRR) and Survival

The role of PMRT in patients with node-positive breast cancer has been heavily studied since the 1990s. The results of a prospective trial from the Danish Breast Cancer Cooperative Group demonstrated a significantly lower frequency of LRR in premenopausal women with stage II–III breast cancer who received PMRT plus chemotherapy compared with those who received chemotherapy alone (9% vs. 32%, *p* < 0.001) [35]. Their subsequent prospective trial, the 82c trial, demonstrated decreased LRR in the PMRT plus tamoxifen cohort compared with the tamoxifen-only cohort (8% vs. 35%, *p* < 0.001) in post-menopausal women with stage II–III breast cancer [36]. Importantly, most of the women in their study were diagnosed with node-positive disease; thus, these initial trials confirmed the utility of PMRT in this patient population. The BEATRICE trial was a phase III randomized clinical trial, including 940 patients that evaluated the efficacy of bevacizumab in patients with triple-negative breast cancer who were treated with mastectomy and systemic therapy [37]. A total of 359 (38.2%) patients enrolled in the trial received PMRT and 581 (61.8%) did not. In patients with N0 disease, no significant increase in locoregional recurrence (LRR)-free 5-year survival was seen with the addition of PMRT (HR = 1.09); however, in patients with N1 disease, patients who underwent PMRT experienced higher 5-year LRR-free survival compared with those who did not receive PMRT (96% vs. 91%, HR = 0.46). Thus, there may be a lower incidence of recurrence with the addition of radiation therapy after a mastectomy in node-negative disease. Similarly, a retrospective analysis of the HERA trial, a phase III randomized clinical trial evaluating the efficacy of trastuzumab in HER-2 positive early-stage breast cancer, found no significant difference in LRR after PMRT in N0 patients (*p* = 0.96) [38]. However, patients in the PMRT cohort with N1-N3 disease experienced improved LRR-free survival compared with those who did not receive PMRT (HR = 0.28, *p* = 0.004).

The impact of molecular subtype on LRR following PMRT has yet to be fully understood. Libson et al. performed a retrospective, single-institution study of 82 stage II breast cancer patients who underwent a mastectomy for a T1/T2 lesion, of which 22 (27%) received PMRT [39]. They found that LRR occurred only in the PMRT cohort, and that time to LRR was significantly lower in the ER-negative group compared with the ER-positive group (64 vs. 82 months, *p* = 0.029). These findings suggest a complex interplay between hormone receptor status and recurrence rates, which may influence clinicians’ and patients’ decisions to use PMRT. Similarly, the results of a cohort study of breast cancer patients by Zhang and colleagues demonstrated that PMRT is significantly associated with lower LRR (HR = 0.42, 95% CI 0.37–0.48) [40]. In their study, hormone receptor status was significantly associated with LRR, with HER2+ patients experiencing a greater risk of LRR (HR = 1.79, 95% CI 1.42–2.26) and ER+/PR+ patients experiencing a lower risk of LRR (HR = 0.82, 95% CI 0.69–1.01). Furthermore, the effect of PMRT on LRR-free survival was the greatest in clinical T2N0–T4N3 patients, suggesting that this patient population may benefit more from PMRT. One cohort study including 884 patients with invasive breast cancer who underwent a mastectomy, including 359 (41.6%) who underwent PMRT, found that patients with triple-negative disease were least likely to receive PMRT (OR = 0.59, 95% CI 0.37–0.93, *p* = 0.02), which may be due to their smaller tumor size and lower N stage [41]. In their study, triple-negative breast cancer had the highest rate of LRR (HR = 5.70, 95% CI 2.92–11.15, *p* < 0.0001). PMRT was not associated with improved LRR in triple-negative cancers (HR = 1.71, 95% CI 0.59–4.92, *p* = 0.32), and there was no association between PMRT and LRR for the overall cohort. These results are consistent with those of Kyndi and colleagues, who showed that triple-negative cancers treated with PMRT were associated with significantly increased LRR (*p* < 0.001) [42].

The efficacy of PMRT in improving overall survival (OS) and disease-free survival (DFS) among breast cancer patients has been demonstrated through various trials and analyses, indicating its potential benefit particularly in certain clinical subgroups. The results of the Danish Breast Cancer Cooperative Group 82b Trial demonstrated an improvement in OS and DFS in patients who received PMRT and chemotherapy compared with chemotherapy alone (OS: 54% vs. 45%, *p* < 0.001; DFS: 48% vs. 34%, *p* < 0.001) [35]. Additionally, their subsequent 82c trial, including post-menopausal women with stage II–III breast cancer, showed improved 10-year OS and DFS in the radiotherapy plus tamoxifen cohort compared with tamoxifen alone (OS: 45% vs. 36%, *p* < 0.001; DFS: 36% vs. 24%, *p* < 0.001) [36]. Wu and colleagues queried the National Cancer Database to assess the impact of PMRT on survival in breast cancer patients with micrometastatic nodal disease [43]. They found a significant association between PMRT and OS in the univariate analysis (HR = 0.75, 95% CI 0.64–0.89). However, this effect was not seen in the multivariate analysis (adjusted HR = 1.01, 95% CI 0.84–1.20), suggesting that demographic and clinical variables may impact the efficacy of PMRT on survival. Zhang and colleagues found that PMRT was associated with significantly improved OS in a multivariate analysis (HR = 0.75, 95% CI 0.65–0.87) and DFS (HR = 0.68, 95% CI 0.58–0.80) [40]. The effect of PMRT on OS was most pronounced in patients with clinical T3, T4, and N0–3 diseases, suggesting that the decision to undergo PMRT may be tailored to this patient population.

## 6. Influence of Post-Mastectomy Radiotherapy on Cosmetic Outcomes and Patient Satisfaction

The decision to undergo adjuvant radiotherapy for patients with implant-based reconstructions remains a challenge, with clinicians needing to weigh the impacts on patient quality of life (QOL) and psychological well-being against the potential for improved outcomes. BREAST-Q is a commonly used survey involving six domains, satisfaction with breasts, overall outcome, process of care, and psychosocial, physical, and sexual well-being, which measures the impact and effectiveness of breast surgery from the patient’s perspective [44].

One appealing factor of IBR after a mastectomy for patients is preserving their image before undergoing additional treatment. While this is a meaningful patient-centered outcome, the extent to which sequencing improves QOL is unclear. One recent study comparing patient satisfaction after receiving either immediate or delayed breast reconstruction found that there was no significant difference in their physical, psychosocial, or sexual well-being when using QOL surveys (median BREAST-Q© and SF-36 scores) after a two-year follow up [45]. On the contrary, patients who received immediate post-mastectomy breast reconstructions rated both their physical and mental health as being significantly better than women who received delayed post-mastectomy breast reconstructions [46].

There are differences in patients’ perceived QOL scores after IBR that are dependent on the style of reconstruction that is performed. In one study that used median BREAST-Q© scores to quantify patient satisfaction, patients who received tissue-expander reconstructions (n = 22) compared to those who received implant-based reconstructions (n = 84) prior to PMRT were not found to have a significant difference in the degree of satisfaction with their breasts (57.2 vs. 56.2) or their perceived physical well-being (73.4 vs. 72.5). A statistically significant difference in perceived psychosocial well-being (72.3 vs. 71.1; *p* < 0.01) and sexual well-being (55.4 vs. 54.0; *p* < 0.01) was observed between the studied groups, although the absolute difference is not likely clinically relevant [13]. Another QOL study retrospectively compared patients who received immediate or delayed implant-based reconstructions to autologous reconstruction inpatients requiring PMRT. Patients with more advanced disease had greater satisfaction with immediate implant-based reconstructions at the expense of increased complications and worse long-term aesthetic scores. Patients with an earlier-stage disease reported improved QOL scores with delayed autologous reconstructions. The need for PMRT decreases patient satisfaction scores regardless of reconstruction modality or timing [47].

These findings reflect those from Kim et. al, which investigated the satisfaction of patients receiving reconstructions either before or after receiving PMRT. While the findings of this study suggest no significant difference between the overall satisfaction of breast reconstructions between patients from both groups, patients who delayed reconstructions until after PMRT were more satisfied with the appearance of their breasts using the Michigan Breast Reconstruction Outcome Study questionnaire (8.3 vs. 7.0; *p* = 0.03) [48]. Lastly, Reinders et al. compared patient satisfaction between those who received either immediate autologous reconstructions or immediate implant-based reconstructions prior to PMRT using the mean BREAST-Q© scores for each group. They found no significant difference between overall patient satisfaction of reconstruction outcome (59.4 vs. 55.8; *p* = 0.421), psychosocial well-being (68.1 vs. 62.5; *p* = 0.152), and physical well-being of the chest (66.9 vs. 62.7; *p* = 0.198), when comparing these groups of patients [49]. However, patients receiving immediate autologous reconstructions before PMRT compared to patients receiving immediate implant-based reconstructions before PMRT were found to have a more significant satisfaction with their breasts (63.7 vs. 50.9; *p* = 0.001) in addition to significantly better perceived sexual well-being (55.5 vs. 46.0; *p* = 0.037) [49]. Overall, the incongruency of the QOL data suggests the optimal integration of reconstruction with PMRT is an personal decision between the provider and patient. Quality of life, life expectancy, surgical complication risk, and desired cosmetic outcomes are important considerations before choosing a reconstruction strategy when PMRT is required for treatment.

To improve patient psychological well-being, a shared decision-making process should be employed that evaluates patient characteristics and potential for adverse psychological or physical complications. A growing body of research suggests that a woman’s decision to undergo breast reconstruction is motivated by a desire to feel normal, while the decision not to have breast reconstruction is related to a fear of complications or the surgery itself [50,51,52]. Thus, the information that women receive regarding their options weighs heavily on the decision to receive reconstructions and PMRT. Future research should focus on implementing shared decision-making processes that thoroughly evaluate patient characteristics and potential risks for adverse psychological or physical complications associated with PMRT and breast reconstruction. This entails setting appropriate expectations for patients regarding the likelihood of complications and the impact on psychological outcomes, while considering factors such as the type of reconstruction and patient risk factors.

## 7. Future Directions and PreMRT

Historically, neoadjuvant radiotherapy (NART) was performed for breast cancers that were considered inoperable prior to the onset of moderate systemic therapy. More recent results of retrospective studies suggest that NART prior to IBR may mitigate the deleterious effects of radiation on reconstruction, however prospective evidence was lacking. A phase 2 single-center RCT published in April 2024 investigated pre-mastectomy radiotherapy (PreMRT) with regional lymph node irradiation (RNI) before autologous IBR [53]. The cohort consisted of 49 patients who received RT to intact breast and regional lymph nodes prior to mastectomy and immediate reconstruction. None of these patients had complete autologous flap losses and 17% of patients suffered skin flap necrosis. There were no local regional recurrences or distant metastasis at follow up (median 23.7 months) [53]. This study met its primary endpoint, and preMRT/RNI was found to be feasible and safe [53]. A larger phase 3 study based on these findings (NCT05774678) will investigate the incidence of adverse events in hypofractionated versus conventionally fractionated PreMRT before immediate autologous reconstructions. The oncologic outcomes of this trial may have practice-changing implications for the sequence of breast cancer treatment, allowing more women to undergo IBR with less risk of reconstruction failure. Over a dozen clinical trials are ongoing in this research space (Table 3).

In cases of triple-negative breast cancer (TNBC) and HER2-positive breast cancer, patients who present pCRs are more likely to have improved survival outcomes, including recurrence-free survival (RFS) and overall survival (OS) [54]. The impact of NART, especially when combined with neoadjuvant chemotherapy (NAC), is still under investigation. Some studies suggest that additional radiotherapy after NAC might benefit patients who do not achieve a pCR, especially in reducing recurrence. However, this benefit appears to be more pronounced in certain subgroups, such as those with residual nodal disease, while others might not see a significant improvement in survival outcomes [55]. Furthermore, the use of neoadjuvant platinum-based chemotherapies has been shown to increase pCR rates in TNBC, although the benefit of long-term survival is still debated. For HER2-positive breast cancer, the dual HER2 blockade with agents like trastuzumab and lapatinib has been associated with higher pCR rates, but the exact role of post-neoadjuvant therapy is still being investigated [54,55].

Overall, the lack of pCRs following neoadjuvant therapy for these breast cancer subtypes necessitates the investigation of alternative high-potency adjuvant treatment strategies to improve patient survival. Unfortunately, the use of more aggressive adjuvant therapy may further complicate the choice of breast reconstruction techniques used to optimize aesthetic outcomes for these patients.

## 8. Conclusions

The controversy surrounding breast reconstruction and radiotherapy reflects greater trends toward prioritizing patient-centered outcomes in oncology. On the one hand, immediate reconstruction after a mastectomy offers practical and psychological benefits that are increasingly supported by the prospective data. However, these benefits may be outdone by the known complications of radiation, including pain, disfiguration, and additional surgery, at the expense of potential disease control. Additionally, adjuvant radiation therapy in the setting of delayed breast reconstruction may be seen as a preferable treatment modality to that of immediate reconstructive techniques, which may place pressure on surgeons to delay definitive reconstructive surgery to avoid complications when the need for radiation is unclear. With continued surgical and medical advancements, we may soon find a treatment sequence that requires less of a compromise between efficacy and quality of life. Until then, we can support patients through shared decision making in a multidisciplinary setting to optimize the sequence of surgery and radiotherapy for each individual.

## Figures and Tables

**Figure 1 cancers-16-02939-f001:**
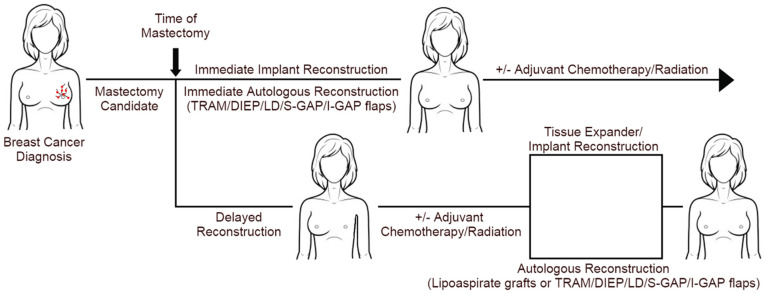
Different modalities of breast reconstruction and PMRT.

**Table 1 cancers-16-02939-t001:** Advantages and disadvantages of different reconstructive techniques.

	Advantages	Disadvantages
Immediate Autologous	Lower chance of capsular contracture Lower morbidity compared with implant One surgery Better QOL vs. delayed options	Often contraindicated due to patient comorbidities or anatomy More invasive than implant Longer recovery time More expensive in the short term
Immediate Implant	One surgery Less complex Shorter recovery time More cost-effective in the short term Better QOL vs. delayed options	Increased risk of complications, including capsular contracture, infection, and skin breakdown
Delayed Autologous	Lower complication and failure rates compared with immediate autologous technique More invasive More cost-effective in the long term	Lower overall QOL Multiple surgeries Less cost-effective in the short term Longer time to achieve cosmetic results Increased risk of capsular contracture
Delayed Expander/Implant	Lower complication and failure rates compared with immediate implant Option to revise implants after TE removal Faster recovery time than delayed autologous options	Lower QOL Multiple surgeries Increased risk of capsular contracture compared with immediate implant

**Table 3 cancers-16-02939-t003:** Current body of clinical trials evaluating breast reconstruction in the setting of PMRT.

NCT	Phase	Number of Patients Enrolled (Estimated *)	Primary Endpoint	Currently Enrolling
NCT05993559	3	1314	Evaluates 5-year survival rate in breast cancer patients receiving a mastectomy and neoadjuvant chemotherapy without PMRT compared to with PMRT	No
NCT05512286	N/A	80	Evaluates patient-reported outcomes for pre-operative and post-mastectomy radiotherapy regarding patients with DIEP flaps	No
NCT05440149	3	1106	Evaluates 7-year survival receiving PMRT/whole breast irradiation versus those not receiving it	Yes
NCT05253170	3	622	Evaluates non-inferiority of complication rates in patients with breast reconstructions between hypofractionated vs. conventional fraction radiotherapy	No
NCT05045287	2	57	Evaluates failure rate of hypofractionated PMRT in patients with two-stage expander/implant reconstructions	Yes
NCT04992650	N/A	50	Evaluates breast skin blood supply after fat grafting in patients with PMRT	By invitation
NCT03523078	N/A	500	Evaluates cosmetic/patient-reported outcomes and complications in patients with and without PMRT	Unknown
NCT03414970	3	897	Evaluates non-inferiority of hypofractionated PMRT complication rates and reoccurrence rates in patients with stage IIa-IIIa breast cancer	No
NCT03319069	3	60	Evaluates efficacy/toxicities of hypofractionated vs. conventional PMRT in high-risk breast cancer patients	Unknown
NCT03072316	N/A	300	Evaluates effects of PMRT on breast cancer patients who received DIEP reconstructions	Unknown
NCT02992574	N/A	1022	Evaluates efficacy/reoccurrence in patients with early high-risk, but node-negative, breast cancer treated with PMRT	Yes
NCT02679040	2	101	Evaluates histological response of patients receiving neoadjuvant chemotherapy and radiation after a mastectomy and immediate reconstruction	No
NCT01925651	N/A	58	Evaluated if bolus usage during PMRT increased the treatment time or decreased efficacy	Complete
NCT01666899	N/A	10	Evaluates effect of PMRT on skin and blood vessels after treatment	Complete
NCT01452672	3	600	Evaluates the necessity of chest wall irradiation alone vs. chest wall and supraclavicular fossa irradiation in PMRT	Unknown
NCT01417286	2	69	Evaluates efficacy and toxicities of accelerated radiotherapy in post-mastectomy patients	Complete
NCT01292772	N/A	12	Evaluates effects of PMRT on patients who had immediate breast reconstructions	Complete
NCT00966888	3	3500	Evaluates efficacy of PMRT in patients with stage II breast cancer compared to observations alone after a mastectomy	Unknown
NCT00005588	3	Not listed	Evaluates different radiation regimens in patients with early-stage breast cancer after a mastectomy	Complete
NCT05483712	N/A	20	Evaluates efficacy of brass mesh bolus compared to the current standard of care in PMRT patients	Yes

*: Estimates based on most recent enrollment numbers listed on trial registry data.

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
