# Peer review of "The Effects of Radiotherapy on the Sequence and Eligibility of Breast Reconstruction: Current Evidence and Controversy"

_cancers, 2024, doi:10.3390/cancers16172939_

Round 1
Reviewer 1 Report
Comments and Suggestions for Authors
The study entitled "The Effects of Radiotherapy on Sequence and Eligibility of Breast Reconstruction: Current Evidence and Controversy" by Campbell et al. is a comprehensive review evaluating immediate breast reconstruction (IBR) and postmastectomy radiotherapy (PMRT) in breast cancer treatment. It shows trends, techniques, and clinical considerations for IBR, such as the use of implants and autologous tissue. The study compares immediate and delayed reconstruction strategies, emphasizing the pros and cons of each in the context of PMRT.
I have some comments:
- Please include keywords;
- Being a review, I believe some important studies are missing. Such as PMID: 36143318 which analyzes immediate breast reconstruction in patients undergoing mastectomy after neo-adjuvant chemotherapy. Please include this study in your review to improve the quality of your manuscript;
- Your conclusions are a bit confusing. Conclusions should be shorter and concise. Please re-write them.
Author Response
- - Please include keywords;
- Thank you for this recommendation. We have added keywords to the cover page of our manuscript [Line 33]
- Being a review, I believe some important studies are missing. Such as PMID: 36143318 which analyzes immediate breast reconstruction in patients undergoing mastectomy after neo-adjuvant chemotherapy. Please include this study in your review to improve the quality of your manuscript;
Thank you for this recommendation. We added language to describe the placement of implant-based reconstruction based on this reference [Lines 88-91]
- Your conclusions are a bit confusing. Conclusions should be shorter and concise. Please re-write them.
Thank you for this recommendation. We have clarified the language of our conclusions section to better explain our thoughts on the current literature [Lines 476-484].
Reviewer 2 Report
Comments and Suggestions for Authors
This is a review on the effects of radiotherapy on the sequence and eligibility of breast reconstruction covering the impact of radiotherapy on reconstruction outcomes, comparing immediate and delayed reconstruction techniques, and discussing complications and long-term results. It summarizes the key clinical studies and evidence, focusing on patient perspectives, quality of life.
Can add a sentence on role of the multidisciplinary team in decision-making.
No other suggestions
Comments on the Quality of English Language
A nice summary well written
Author Response
- Can add a sentence on role of the multidisciplinary team in decision-making.
Thank you for this recommendation. We have added language to emphasize the role of a multidisciplinary team in optimizing the sequence of care for patients considering post-mastectomy breast reconstructive surgery [Line 484].
Reviewer 3 Report
Comments and Suggestions for Authors
The issue of breast reconstruction and radiotherapy is one of big interest as it requires a balance between assuring an improved quality of life and avoinding surgical complications.
The authors offer a narrative review that discusses the effects of radiotherapy on the sequencing and eligibility for breast reconstruction post-mastectomy, focusing on immediate versus delayed reconstruction.
The article is well-structured. The introduction establishes the context and relevance of the topic. The subjects are addressed in a logical manner, from introducing reconstruction techniques to discussing the impact of radiotherapy on these options, number and possible complications.
The discussion of each reconstruction method provides an informative overview. However a systematic review would be more scientifically significant in this manner.
No keywords are offered by the authors.
They are some editing improvements that need to be done (eg. upper letter case in titles, number of tables - they are 3 Tables 3 in the text and this creates confusion when reading the manuscript)
Also when presenting the data in Table 3 (pg 6) the authors can choose a more efficient form in which the data is not repetitive.
In the conclusion section, the authors added new information not discussed in the body of the manuscript about neoadjuvant radiotherapy.
Author Response
- No keywords are offered by the authors.
- Thank you for this recommendation. We have added keywords to the cover page of our manuscript [Line 33]
- They are some editing improvements that need to be done (eg. upper letter case in titles, number of tables - they are 3 Tables 3 in the text and this creates confusion when reading the manuscript)
Thank you for this recommendation. We have corrected the table numbers to be numerically correct, and we have corrected several formatting and grammatical errors throughout the manuscript.
- Also when presenting the data in Table 3 (pg 6) the authors can choose a more efficient form in which the data is not repetitive.
- Thank you for this recommendation. We believe that this table is structured in such a way to represent the cumulative sum of investigative works on this topic. Additionally, the table presents a more complete overview of the data discussed in the manuscript, while not burdening the reader with unnecessary text in the body paragraph. While it may appear repetitive, we believe that it is important to represent these ongoing studies in such a comprehensive manner.
- In the conclusion section, the authors added new information not discussed in the body of the manuscript about neoadjuvant radiotherapy.
Thank you for this recommendation. We have clarified the language surrounding this concern as to not introduce new information at the end of the manuscript, but to instead draw attention to competing decisions surgeons may face in choosing IBR vs. DBR [Lines 477-481].